# Real-Time Remaining Useful Life Prediction of Cutting Tools Using Sparse Augmented Lagrangian Analysis and Gaussian Process Regression

**DOI:** 10.3390/s23010413

**Published:** 2022-12-30

**Authors:** Xiao Qin, Weizhi Huang, Xuefei Wang, Zezhi Tang, Zepeng Liu

**Affiliations:** 1Adam Smith Business School, University of Glasgow, Glasgow G12 8QQ, UK; 2Department of Physics and Astronomy, University College London, London WC1E 6BT, UK; 3School of Engineering, University of Manchester, Manchester M13 9PL, UK; 4Department of Automatic Control and Systems Engineering, University of Sheffield, Sheffield S10 2TN, UK

**Keywords:** remaining useful life estimation, cutting tools, advanced manufacturing, sparse augmented lagrangian, gaussian process regression

## Abstract

Remaining useful life (RUL) of cutting tools is concerned with cutting tool operational status prediction and damage prognosis. Most RUL prediction methods utilized different features collected from different sensors to predict the life of the tool. To increase the prediction accuracy, it is often necessary to mount a great deal of sensors on the machine in order to collect more types of signals, which can heavily increase the cost in industrial applications. To deal with this issue, this study, for the first time, proposed a new feature network dictionary, which can enlarge the number of candidate features under limited sensor conditions, and the developed dictionary can potentially contain as much useful information as possible. This process can replace the installation of more sensors and incorporate more information. Then, the sparse augmented Lagrangian (SAL) feature selection method is proposed to reduce the number of candidate features and select the most significant features. Finally, the selected features are input to the Gaussian Process Regression (GPR) model for the RUL estimation. Extensive experiments demonstrate that our proposed RUL estimation framework output performs traditional methods, especially for the cost savings for on-line RUL estimation.

## 1. Introduction

In recent years, the fourth industrial revolution (Industry 4.0) has brought about an increase in demand for precision machining and production manufacturing. As one of the typical manufacturing machines, high-speed computer numerically controlled (CNC) milling machines are widely used in industry [1]. Statistics show that CNC milling machines now account for approximately 46% or more of the total modern industrial machining cluster [2]. In order to ensure the safe operation of CNC machines and the quality of the products, cutting tools have a crucial role in the overall CNC machine system. However, cutting tools are usually operated in high-pressure and harsh environments, which can result in wear and tear on the tools. Some of the studies show that the failure rate of cutting tools is higher than 38% among all mechanical failures of CNC machines. When tool wear exceeds a certain threshold, the machined part no longer meets the machining requirements, which in turn causes losses to the plant. Drawing on these insights, there is an urgent need for researchers to develop efficient and cost-effective approaches to estimate the real-time remaining useful life (RUL) of tools in order to free up sufficient time to plan replacements and repairs, which can prevent potential unscheduled machine downtime and shorten the manufacturing cycle time of products [3,4].

At present, a variety of active achievements have been made to estimate the RUL predictions of cutting tools including direct and indirect techniques [5]. Direct methods basically use machine vision systems to capture the images of the cutting cools by means of digital cameras or optical sensors [6,7,8]. For example, Prasad et al. [9] used a pair of stereo images taken from two different locations to measure the depth of wear. Then the measured wear was used as test data to predict flank wear using a back propagation neural network. Lanzetta [10] measured and classified various categories of wear (e.g., flank wear, crater wear, fracture, and breakage) of cutting tool inserts using a vision sensor approach. The approach comprised a camera, an autofocus zoom lens, and different types of illumination. A resolution enhancement algorithm was used by the author which can enhance the resolution of the measured picture to 40 μm/pixel. The advantages of direct methods are that they can provide high identification accuracy for the tool wear measurement. However, they are essentially off-line techniques that often result in unnecessary machine downtime and increased costs due to lost productivity. To address this issue, indirect methods have been extensively investigated which are conducted by measuring parameters that are related to the condition of the cutting tool, i.e., acoustic, vibration, force, and sound [11,12,13]. Compared with direct approaches, indirect methods facilitate the processing of different types of signals and can easily be applied to on-line monitoring, and, therefore, have been a popular topic for estimating the RUL of cutting tools.

The application of indirect methods is based on two stages containing feature extraction and feature-based RUL estimation [1]. In regard to feature extraction, the significant features are extracted from the candidate feature dictionary developed from the collected signals in order to represent the status of the cutting tool. With respect to the feature-based RUL estimation, the relationship between the extracted features and the status of the cutting tool is established to predict the life of the cutting tool. For example, Liu et al. [14]. utilized 14 signal features developed from the collected acoustic emission (AE) signals during the feature extraction stage and input the features to the support vector regression (SVR) for the life prediction during the RUL estimation stage. The results show that the method can achieve a prediction accuracy of up to 94.35%. Hu et al. [15] extracted time-domain features, such as mean, variance, and kurtosis; frequency-domain features, such as spectral kurtosis and spectral skewness; and time-frequency domain features, such as wavelet energy as the significant features; and the Long Short-term Memory (LSTM) network was applied to conduct tool wear prediction. Li et al. [16] utilized 16 features from the force signals during the feature extraction stage and applied a fuzzy neural network for the RUL estimating stage. Wang et al. [17] proposed a multisensory data fusion approach for the feature extraction and SVR for the RUL estimation. As can be seen from the aforementioned literature, to deliver the desired RUL estimation performance of indirect methods, the feature extraction that converts the raw signal into useful knowledge about the health status of cutting tools plays a fundamental role [1]. To extract significant features, the size of the candidate feature dictionary is required large enough to contain sufficient information about the tool state. Nevertheless, this process gives rise to a significant challenge which requires more sensors to be fitted on the machine which can heavily increase the cost.

To solve this issue, in the present study, a new feature network dictionary is proposed during the off-line training stage that enlarges the number of candidate features under limited sensor conditions, and the developed candidate features in the dictionary can potentially contain as much useful information as possible. After that, the sparse augmented Lagrangian (SAL) feature selection method is proposed during the feature extraction stage which can reduce the number of candidate features and select the most significant features [18]. The characteristics of SAL are that it utilizes Split Augmented Lagrangian Shrinkage Algorithm (SALSA) to produce some intermediate features with a sub-sampling technique, and then only the features with high selecting probability are chosen as the final selected features. Therefore, SAL has a very high sparsity and only a very few significant features can be selected. These selected features are often associated with some specific sensors, and only these specific sensors will be used for on-line monitoring. The sensors that are not selected can be removed from the machine, which can save costs for on-line monitoring. Finally, the selected features are input to the Gaussian Process Regression (GPR) model for the RUL estimation. GPR is a non-parametric Bayesian regression method that performs well on small datasets and has the capability to produce uncertainty measurements on the predictions [19]. Promising results have demonstrated the effectiveness of the newly proposed RUL estimation framework including SAL and GPR has high prediction accuracy, demonstrating the advantage over many existing methods.

The main contributions of this article can therefore be summarized as follows.

First, the feature network dictionary is proposed to extend the number of candidate features under limited sensor conditions in order to contain as much useful information as possible.Second, a SAL feature selection approach is proposed. It has very high sparsity and only the significant features can be selected.Finally, during the on-line monitoring, only the selected sensors will be used, and the sensors that are not selected can be potentially removed from the machine creating a favorable opportunity for cost savings in real-world industry applications.

The remainder of this paper is organized as follows. In Section 2, the theoretical background of the feature network dictionary, SALSA, SAL for feature selection, and GPR are discussed in detail. Then, Section 3 is dedicated to a description of the first case study and Section 4 shows the results of the second case study. Section 5 discusses the proposed method in detail. Finally, Section 6 concludes this study.

## 2. Theoretical Background

### 2.1. Feature Network Dictionary

During the machining process, different types of sensors are installed on the machine to monitor the cutting tool status. Suppose $x_{i,j}\left(t\right)$, t=1,...,T, i=1,..,I and j=1,...,J is the measured raw signal responses from the *i*-th sensor output during the *j*-th cutting process, under which the tool wear is denoted as $y_{j}$. The signal features of $x_{i,j}\left(t\right)$ can be used to assess the tool wear severity and are given by
(1)$$P_{i,j}=\left[P_{i,j}^{1},P_{i,j}^{2},...,P_{i,j}^{n}\right],n=1,...,N$$
and
(2)$$P_{i,j}^{n}=f_{i,j}^{n}\left(x_{i,j}\left(t\right)\right)$$
where the notation *n* represents the *n*-th signal feature and *N* represents the number of signal features for *i*-th sensor output. $P_{i,j}^{n}$ is the generated feature and $P_{i,j}$ represents the feature matrix enclosing *N* signal features. If the features from all *I* sensor outputs are combined in a single matrix, the signal feature dictionary during the *j*-th cutting process is given by [20]
(3)$$\begin{array}{cc} P_{j}= & \left[P_{1,j},P_{2,j},...,P_{I,j}\right]\\ = & [P_{1,j}^{1},...,P_{1,j}^{N},P_{2,j}^{1},...,P_{2,j}^{N},...,P_{I,j}^{1},...,P_{I,j}^{N}],j=1,...,J \end{array}$$

From (Equation 3), there are a total of NI features in the signal feature dictionary. To extend the size of the dictionary under the *I* sensor outputs conditions in order to contain as much useful information as possible, the feature network dictionary is developed where the process of the extension of (Equation 3) is an iterative process. Support $P_{j}^{l}$ is the extended dictionary after the *l*-th extension iteration, and the original feature dictionary (Equation 3) can be written as
(4)$$\begin{array}{cc} P_{j}^{0}=P_{j}= & \left[P_{1,j}^{1},...,P_{1,j}^{N},P_{2,j}^{1},...,P_{2,j}^{N},...,P_{I,j}^{1},...,P_{I,j}^{N}\right]\\ = & [a_{1},a_{2},\cdots ,a_{\phi }],\phi =1,\cdots ,NI \end{array}$$

Define $A^{1}$ as an intermediate matrix during the *l*-th iteration, which records the result of the matrix product, providing
(5)$$A^{1}={P_{j}^{0}}^{T}\times P_{j}^{0}=\left[\begin{array}{cccc} {\left(a_{1}\right)}^{2} & a_{1}a_{2} & \cdots & a_{1}a_{\phi }\\ a_{2}a_{1} & {\left(a_{2}\right)}^{2} & \cdots & a_{2}a_{\phi }\\ \vdots & \vdots & \ddots & \vdots \\ a_{\phi }a_{1} & a_{\phi }a_{1} & \cdots & {\left(a_{\phi }\right)}^{2} \end{array}\right],$$

The vectorization of the matrix $A^{1}$, denoted $\mathrm{VEC}(A^{1})$, is the $1\times \phi^{2}$ row vector: (6)$$\mathrm{VEC}\left(A^{1}\right)=\left[{\left(a_{1}\right)}^{2},a_{1}a_{2},\cdots ,a_{1}a_{\phi },a_{2}a_{1},{\left(a_{2}\right)}^{2},\cdots ,a_{2}a_{\phi },\cdots ,a_{\phi }a_{1},a_{\phi }a_{1},\cdots ,{\left(a_{\phi }\right)}^{2}\right]$$

Define D as a process to delete the element which has the same expression of a vector, so $D(\mathrm{VEC}\left(A^{1}\right))$ can be expressed as
(7)$$D\left(\mathrm{VEC}\left(A^{1}\right)\right)=\left[{\left(a_{1}\right)}^{2},a_{1}a_{2},\cdots ,a_{1}a_{\phi },{\left(a_{2}\right)}^{2},\cdots ,a_{2}a_{\phi },\cdots ,{\left(a_{\phi }\right)}^{2}\right]$$
where one of the same expression elements of $A^{1}$, such as $a_{2}a_{1}$ and $a_{1}a_{2}$, will be deleted. Based on the permutations and combinations formula [21], the number of elements in $D(\mathrm{VEC}\left(A^{1}\right))$ is $\frac{NI(NI+1)}{2!}$.

Then, the extended dictionary after the first iteration is
(8)$$\begin{array}{cc} P_{j}^{1} & =[P_{j}^{0},D\left(VEC\left(A^{1}\right)\right)]\\ \; & =[\underset{1\times NI}{\underset{︸}{a_{1},a_{2},\ldots ,a_{\phi }}},\underset{1\times \frac{NI(NI+1)}{2!}}{\underset{︸}{{\left(a_{1}\right)}^{2},a_{1}a_{2},\ldots ,a_{1}a_{\phi },{\left(a_{2}\right)}^{2},\ldots ,a_{2}a_{\phi },\ldots ,{\left(a_{\phi }\right)}^{2}}}] \end{array}$$
where the size of the dictionary $P_{j}^{1}$ extends to $1\times (NI+\frac{NI(NI+1)}{2!})$.

In the same way, after the *l*-th extension with l>0 and $l\in Z^{+}$, the extended dictionary can be expressed as
(9)$$P_{j}^{l}=\left[P_{j}^{0},D\left(\mathrm{VEC}\left(A^{l}\right)\right)\right]$$
where
(10)$$A^{l}={P_{j}^{0}}^{T}\times P_{j}^{l-1}$$
and the dictionary $P_{j}^{l}$ is defined as the feature network dictionary during the *j*-th cutting process having the size of 1×(NI+(NI+12)+(NI+23)+⋯+(NI+ll+1)) where (NI+ll+1)=(NI+l)!(l+1)!(NI−1)!. To facilitate understanding of the process of feature network dictionary generation, Figure 1 further demonstrates the procedure.

As a consequence, after a total of *J* cutting processes, the feature network dictionary can be rewritten as
(11)$$P={\left[P_{1}^{l},...,P_{J}^{l}\right]}^{T}$$
where the feature network dictionary P is a J×(NI+(NI+12)+(NI+23)+⋯+(NI+ll+1)) matrix.

### 2.2. Split Augmented Lagrangian Shrinkage Algorithm (SALSA)

Assume that the tool wear vector is $y={\left[y_{1},...,y_{J}\right]}^{T}$, then the tool wear estimation model can be expressed as
(12)y=PΘ+Ξ
where Θ stands for the model coefficients, and $\Xi ={\left[\xi_{1},...,\xi_{J}\right]}^{T}$ is the unmodeled noise.

The feature network dictionary is required to be large enough to contain sufficient information with the aim of giving an accurate estimation of the tool wear. However, the features in the dictionary P are usually redundant and may not be necessary for the tool wear estimation. A sparse model representation is therefore beneficial in this case [22,23].

The sparse solution of y=PΘ+Ξ can be obtained by solving the following $l_{1}$-norm optimization problem [24]:(13)$$\Theta =\arg\min_{\Theta }\left\{\frac{1}{2}{∥P\Theta -y∥}_{2}^{2}+\lambda {∥\Theta ∥}_{1}\right\}$$

SALSA is applied to solve this equation, which is a variable splitting technique converting the original $l_{1}$-norm minimization problem into a series of subproblems which can be solved separately. Equation (Equation 13) can be therefore converted as:(14)$$\begin{array}{cc} \min_{\Theta ,v\in R^{M}}\; & f_{1}\left(\Theta \right)+f_{2}\left(v\right)+\frac{\mu }{2}{∥\Theta -v∥}_{2}^{2}\\ s.t.\; & \;\;\;v-\Theta =0 \end{array}$$
where $f_{1}\left(\Theta \right)=\frac{1}{2}{∥P\Theta -y∥}_{2}^{2},f_{2}\left(v\right)=\lambda {∥v∥}_{1}$ and μ is the Lagrange multiplier. The solution of (Equation 14) can be approximated to the weighted $l_{1}$-norm optimization problem (Equation 13) with the increase of μ. By applying the augmented Lagrangian method, the optimization problem of (Equation 14) is then expressed as:(15)$$\begin{array}{cc} L_{\mu }\left(\Theta ,v,u\right)= & f_{1}\left(\Theta \right)+f_{2}\left(v\right)-u^{T}\left(\Theta -v\right)\\ \; & +\frac{\mu }{2}{∥\Theta -v∥}_{2}^{2} \end{array}$$
where *u* is a vector of Lagrange multipliers. Substitute d=u/μ into (Equation 15), then the problem is transformed into:(16)$$L_{\mu }\left(\Theta ,v,d\right)=f_{1}\left(\Theta \right)+f_{2}\left(v\right)+\frac{\mu }{2}{∥\Theta -v-d∥}_{2}^{2}$$

The problem (Equation 16) is then solved by converting it into three suboptimization problems: solving Θ, *u* and *d* individually, and in detail:(17)$${\hat{\Theta }}_{k+1}={\left(P^{T}P+\mu I\right)}^{-1}\left(P^{T}y+\mu \left(v_{k}+d_{k}\right)\right)$$
(18)$$\begin{array}{cc} v_{k+1}= & \max\left(0,\left({\hat{\Theta }}_{k+1}-d_{k}\right)-\mu /\lambda \right)\\ \; & -\max\left(0,-\left({\hat{\Theta }}_{k+1}-d_{k}\right)-\mu /\lambda \right) \end{array}$$
(19)$$d_{k+1}=d_{k}-\left({\hat{\Theta }}_{k+1}-v_{k+1}\right)$$
where ${\hat{\Theta }}_{k+1}$ represents the estimation of Θ and λ is the penalty parameter. It can be seen that ${\hat{\Theta }}_{k+1}$ is calculated iteratively and the iteration will stop when the nonzero elements of ${\hat{\Theta }}_{k+1}$ and ${\hat{\Theta }}_{k}$ have the same sign and location. Algorithm 1 briefly summarizes the main procedure of the SALSA algorithm.
**Algorithm 1** SALSA Algorithm**Input:** Model signal **y** and dictionary **P****Initialization:** sets k=0, the Lagrange multiplier μ=0.1, penalty parameter $\lambda =1\times 10^{-3}$, $v_{0}=d_{0}=0$   1:   **while** sign ${\hat{\Theta }}_{k+1}={\hat{\Theta }}_{k}$, $loc_{k+1}=loc_{k}$
**do**   2:       ${\hat{\Theta }}_{k+1}={\left(P^{T}P+\mu I\right)}^{-1}\left(P^{T}y+\mu \left(v_{k}+d_{k}\right)\right)$   3:       $v_{k+1}=\max\left(0,\left({\hat{\Theta }}_{k+1}-d_{k}\right)-\mu /\lambda \right)-\max\left(0,-\left({\hat{\Theta }}_{k+1}-d_{k}\right)-\mu /\lambda \right)$   4:       $d_{k+1}=d_{k}-\left({\hat{\Theta }}_{k+1}-v_{k+1}\right)$   5:       k←k+1   6:   **end while**

### 2.3. Sparse Augmented Lagrangian (SAL) Algorithm for Feature Selection

By applying the SALSA algorithm, the $l_{1}$-norm optimization problem can be solved. However, the results are often not sufficiently sparse. To solve this issue, the SAL algorithm is used to improve the performance of SALSA. Firstly, SAL uses SALSA to generate a series of intermediate models. Secondly, certain features from the model series are selected to build the final model. The detailed procedure of the SAL algorithm is presented as follows:(1)The downsampling method is repeatedly used to process the tool wear vector y and the feature network dictionary P. Specifically, some rows are randomly selected from y to form a new vector as $y_{\upsilon }$. Similarly, the same rows are also selected from P to form a new matrix denoted as $P_{\upsilon }$. After that, the intermediate models based on $y_{\upsilon }$ and $P_{\upsilon }$ can be estimated as
(20)$$y_{\upsilon }=P_{\upsilon }\Theta_{\upsilon }+\Xi_{\upsilon }$$
where υ indicates the υ-th repetition of the downsampling method with υ=1,...,Υ, and $y_{\upsilon }\in R^{J_{s}}$, Pυ∈RJs×(NI+(NI+12)+(NI+23)+⋯+(NI+ll+1)) and
(21)$$J_{s}=\kappa \times J,0<\kappa <1$$
where $J_{s}$ is the number of random subsampling, and κ represents the ratio of the downsampling.(2)SALSA is then applied to calculate $\Theta_{\upsilon }$ from the υ-th intermediate model using the downsampling data $y_{\upsilon }$ and $P_{\upsilon }$.(3)The set of the selected features by SALSA for the υ-th intermediate model can be expressed as
(22)Mυ=pξ|Θξ≠0,ξ=1,...,NI+NI+12+NI+23+⋯+NI+ll+1
where $p^{\xi }\in P$ and $\Theta^{\xi }\in \Theta$. The notation ξ is the $\xi_{th}$ element of the Θ, and $p^{\xi }$ are the $\xi_{th}$ features.(4)Based on all $M_{\upsilon }$, υ=1,...,Υ, the frequency of each feature being selected in all sets can be calculated. Define selection frequency (SF) as
(23)$$\eta \left(p_{\xi }\right)=\chi \left(p_{\xi }\right)/\Upsilon$$
where χ and η are the select times and the select frequency, respectively, of $p_{\xi }$.(5)Then, the newly selected dictionary $P_{s}$ is defined as
(24)$$P_{s}=\left\{p_{\xi }|\eta \left(p_{\xi }\right)\geq \delta \right\}$$
where the notation δ indicates a predefined feature selection threshold. This method can discard certain irrelevant features and only keep the features with high SF. The newly selected dictionary can be expressed as
(25)Ps=p1,p2,...,pϑ,ϑ≤NI+NI+12+NI+23+⋯+NI+ll+1(6)Finally, the location of each selected feature in the feature network dictionary P is recorded as Λ.

The final implementation of SAL for feature selection can be summarized in Algorithm 2 and Figure 2.
**Algorithm 2** SAL for feature selction**Input:** Model signal y, dictionary **P**, downsampling ratio κ and feature selection threshold δ**Output:** Selected features $P_{s}$ and the location Λ   1:   **for**
υ=1:Υ
**do**   2:       Random downsampling →$y_{\upsilon }$ and $P_{\upsilon }$   3:       SALSA algorithm in **Algorithm** 1   4:       $M_{\upsilon }=\left\{p^{\xi }|\Theta^{\xi }\neq 0\right\}$   5:       $\eta \left(p_{\xi }\right)=\chi \left(p_{\xi }\right)/\Upsilon$   6:       $P_{s}=\left\{p_{\xi }|\eta \left(p_{\xi }\right)\geq \delta \right\}=\left[p_{1},p_{2},...,p_{\vartheta }\right]$   7:       The location of each selected feature is denoted as Λ.   8:   **end for**

### 2.4. Gaussian Process Regression (GPR) for the Remain Useful Life (RUL) Estimation

After the features are selected by SAL, GPR is applied to conduct the RUL estimation of the cutting tool. Assume the selected features $P_{s}$ and the measured tool wear vector y can be divided into two datasets which are the training datasets used for GPR model training and the validation/testing dataset used for model evaluation. The training datasets are denoted as $P_{s}^{\alpha }$ and $y^{\alpha }$, respectively, where $P_{s}^{\alpha }$ indicates the selected dictionary and $y^{\alpha }$ indicates the measured tool wear vector. Similarly, the validation/testing datasets are denoted as $P_{s}^{\beta }$ and $y^{\beta }$, respectively.

The first step for the RUL estimation is to define a Gaussian kernel by using the training datasets. In the present study, the radial basis function (RBF) is selected as the kernel function because it has outstanding nonlinear ability and infinitely differentiable characteristics [19]. The RBF kernel is defined as [25]
(26)$$K\left(P_{s}^{\alpha },P_{s}^{\alpha }\right)=\sigma_{f}^{2}\exp(-\frac{1}{2l}∥P_{s}^{\alpha }-P_{s}^{\alpha }∥）$$
where $\sigma_{f}$ and *l* are hyperparameters.

After that, the predictive mean of the RUL of the validation/testing dataset denoted as ${\bar{y}}^{\beta }$ can be derived
(27)$${\bar{y}}^{\beta }=K{\left(P_{s}^{\alpha },P_{s}^{\beta }\right)}^{T}L^{T}\backslash \left(L\backslash y\right)$$
with
(28)$$L=C(K\left(P_{s}^{\alpha },P_{s}^{\alpha }\right)+\sigma_{y}^{2}I)$$
where
(29)$$K\left(P_{s}^{\alpha },P_{s}^{\beta }\right)=\sigma_{f}^{2}\exp\left(-\frac{1}{2l}∥P_{s}^{\alpha }-P_{s}^{\beta }∥\right)$$

The function C represents the Cholesky decomposition, and the notation $\sigma_{y}$ presents the noise estimation parameter relating to the noise level of y.

Then, the predictive variance of the RUL of the validation/testing dataset is represented as
(30)$$V\left(y^{\beta }\right)=K\left(P_{s}^{\beta },P_{s}^{\beta }\right)-v^{T}v,v=L\backslash K{\left(P_{s}^{\alpha },P_{s}^{\beta }\right)}^{T}$$
where V(.) indicates the variance function.

Finally, the log marginal likelihood can be calculated as
(31)$$\log p\left(y|P_{s}^{\alpha }\right)=-\frac{1}{2}y^{T}\Gamma -\sum_{i}\log L_{ii}-\frac{J}{2}\log2\pi$$
where the notation $\Gamma =L^{T}\backslash \left(L\backslash y\right)$.

### 2.5. RUL Estimation of the Cutting Tool

The cutting tool RUL estimation can be performed using the following algorithm, which has two parts, including off-line training and on-line RUL estimation.

Firstly, off-line training will extract the significant features from the proposed feature network dictionary, and train the GPR model off-line. The detailed process is as follows:(1)The collected raw signals from different sensor outputs are used to generate a feature network dictionary denoted as P, and the features inside the dictionary P are normalized by using min-max normalization.(2)The normalized feature network dictionary P and the measured tool wear vector y are divided into two parts which are the training datasets and validation datasets. The training datasets are denoted as $P^{\alpha }$ and $y^{\alpha }$, respectively, and the validation datasets are denoted as $P^{\beta }$ and $y^{\beta }$, respectively.(3)The generated dictionary $P^{\alpha }$ and the measured tool wear vector $y^{\alpha }$ are used to train the SAL feature selection model, and the selected dictionary is denoted as $P_{s}^{\alpha }$, where the location of each selected feature in the dictionary $P^{\alpha }$ is recorded as Λ. Based on Λ, the significant features from the validation dataset denoted as $P_{s}^{\beta }$ can be directly selected.(4)GPR is conducted to estimate the RUL of the cutting tools. First, $P_{s}^{\alpha }$ is applied to define the Gaussian kernel. Then, the predictive mean denoted as ${\bar{y}}_{s}^{\beta }$ and the predictive variance denote as $V(y^{\beta })$ of the validation dataset can be estimated.(5)Mean square error (MSE) shown below is used to evaluate the predictive results.
(32)$${\mathrm{MSE}}^{\beta }=\frac{1}{\varsigma }{∥y_{s}^{\beta }-{\bar{y}}_{s}^{\beta }∥}^{2}$$
where the notation ς indicates the length of $y_{s}^{\beta }$. If the ${\mathrm{MSE}}^{\beta }$ is greater than a predefined threshold $T_{\beta }$, the corresponding parameters of the SAL feature selection procedure need to be re-tuned, and (4) and (5) need to be repeated until ${\mathrm{MSE}}^{\beta }\leq T_{\beta }$.

The second part is the on-line RUL estimation using the on-line extracted features and off-line trained SAL and GPR models to conduct the on-line RUL prediction. The procedure is as follows:(1)Based on the location Λ, the significant features can be directly selected from the on-line collected signals.(2)The selected features are input to the off-line trained GPR model for the on-line RUL estimation.

Figure 3 presents a schematic flowchart of the proposed off-line training and on-line RUL estimation.

## 3. Case Study 1: PHM Data Challenge Datasets

### 3.1. Experiment Setup

To demonstrate the effectiveness of the proposed SAL and GPR framework, the down milling operation experiment conducted by the PHM society in 2010 is employed as shown in Figure 4 [26]. As can be seen, different types of sensors including the dynamometer, accelerometer, and acoustic emission (AE) sensor are used to mount on the different places of the machine with the aim of collecting sensor signals to conduct the RUL of the cutting tools. First, a Kistler quartz 3-component dynamometer is installed between the workpiece and the machining table in order to measure the cutting forces in X, Y, and Z directions during machining. The collected signals are denoted as ${\mathrm{force}}_{x}$, ${\mathrm{force}}_{y}$, and ${\mathrm{force}}_{z}$, respectively. Then, three Kistler piezo accelerometers denoted as ${\mathrm{acce}}_{x}$, ${\mathrm{acce}}_{y}$, and ${\mathrm{acce}}_{z}$, respectively, are mounted in the X, Y, and Z directions of the workpiece in order to measure the vibrations in each of these directions. Furthermore, a Kistler AE sensor denoted as ${\mathrm{AE}}_{\mathrm{RMS}}$ is used, which can measure high-frequency energy signals generated during material removal from the workpiece in the machining process. As a consequence, seven sensor outputs which are ${\mathrm{force}}_{x}$, ${\mathrm{force}}_{y}$, ${\mathrm{force}}_{z}$, ${\mathrm{acce}}_{x}$, ${\mathrm{acce}}_{y}$, ${\mathrm{acce}}_{z}$, and ${\mathrm{AE}}_{\mathrm{RMS}}$, respectively, are collected from the applied sensors, and these sensor outputs are sampled by a NI DAQ PCI 1200 board with 12 kHz sampling rate.

During the experiment, the cutting tool with three flute cutters was utilized to cut the workpiece in order to machine a sloping surface [16]. The cutting speed is set up to 4.7 m/min and the spindle speed is set to 23,600 rpm. After each cutting process, the machine will stop to measure the flank wear of the cutter, which was conducted by employing the LEICA MZ12 microscopy system. At the end of the experiment, a total of three cutting tools denoted as T1, T2, and T3, respectively, were used to machine three aluminium workpieces, and each cutting tool made a total of 300 cutting processes so as to produce 300 datasets enclosing 7 columns sensor outputs and 1 column tool wear measurements.

### 3.2. Off-Line Training and Validation

#### 3.2.1. Sparse Augmented Lagrangian (SAL) Algorithm for the Feature Selection

To conduct the SAL-based feature selection, as can be seen in Figure 5, the original datasets collected from T1 and T2 were first used as the training and validation datasets and the rest dataset collected from T3 was used as the testing dataset. As shown in Table 1, this experimental design for data partitioning is denoted as E1 where the training and validation datasets contain 600 (=300×2) datasets and the testing dataset contains 300 datasets.

During the off-line training and validation stage, 90% of the training and validation datasets (540 datasets) were randomly selected as the training dataset and the rest 10% of the datasets (60 datasets) were used as the validation dataset. Then, the training datasets were used to generate a feature network dictionary, and the features utilized in the current study are presented in Table 2. Based on the utilized features, a feature dictionary can be generated according to Equation (Equation 11). As can be seen in the equation, the critical parameter for the feature network dictionary is the extension iteration *l*. The ideal value of *l* can help in producing the gratifying size of the feature dictionary to contain as much useful information as possible. However, the increase of *l* will greatly expand the size of the initial dictionary matrix which heavily increases the computational load. For the current study, the initial extension iteration *l* is set to 1 for the SAL-based feature selection. According to Equation (Equation 11), the feature network dictionary of the training dataset is a 540 × 5670 matrix, meaning that the number of the designed features in the dictionary is extended to 5670. In a similar way, the validation dataset can also be used to generate a feature network dictionary where the size of the dictionary is 60 × 5670.

After the feature dictionary is generated, the SAL algorithm was applied to select the significant features from the candidate 5670 features. Based on the algorithm introduced in Section 2.3, only 8 features denoted as $P_{s}^{\alpha }$ were finally selected among the 5670 candidate features, where the selected features are ${\mathrm{AE}}_{\mathrm{RMS}}$, ${\mathrm{force}}_{x}\times {\mathrm{force}}_{x}$, ${\mathrm{force}}_{x}\times {\mathrm{force}}_{y}$, ${\mathrm{force}}_{x}\times {\mathrm{force}}_{z}$, ${\mathrm{force}}_{y}\times {\mathrm{force}}_{y}$, ${\mathrm{force}}_{y}\times {\mathrm{force}}_{z}$, ${\mathrm{force}}_{z}\times {\mathrm{force}}_{z}$, and ${\mathrm{AE}}_{\mathrm{RMS}}\times {\mathrm{force}}_{z}$. The location of these selected features is recorded as Λ. As can be seen from the selected features, they are only related to the AE signal and the force signals, and there is no dependence on the vibration signals. Therefore, for the future on-line RUL estimation, there will be no need to install vibration sensors, which is very promising for industrial applications because the combination of feature network dictionary and SAL can be effective in cost saving. Finally, according to the location Λ of the selected features in the dictionary, the selected features of the validation dataset can be represented as $P_{s}^{\Lambda }$.

#### 3.2.2. Gaussian Process Regression (GPR) for the RUL Estimation

After the features were selected via the SAL algorithm, they were then input to GPR for the RUL estimation. First, based on the algorithm introduced in Section 2.4, 8 selected features $P_{s}^{\alpha }$ from the training dataset were used to train the GPR model. After that, the 8 selected features $P_{s}^{\beta }$ from the validation dataset were input to the trained GPR model in order to predict the RUL of the validation dataset. To estimate the performance of the prediction results, the mean square error (MSE) and the area of uncertainty (AoU) are used. The equation of the MSE is computed as
(33)$$\mathrm{MSE}=\frac{1}{\varsigma }\sum_{\varsigma =1}^{\Xi }(y_{\varsigma }^{v}-{\hat{y}}_{\varsigma }^{v}{)}^{2}$$
where $y_{\varsigma }^{\beta }$ and ${\hat{y}}_{\varsigma }^{\beta }$ are the measured tool wear of the validation dataset and the GPR predicted tool wear, respectively. The notation Ξ indicates the length of the predicted tool wear ${\hat{y}}_{\varsigma }^{\beta }$, and ς indicates the index of the dataset. The smaller the MSE, the more accurate the model’s predictions will be. Furthermore, the area of uncertainty (AoU) is proposed to measure the uncertainty of the prediction results, which can be expressed as
(34)$$\mathrm{AoU}=\frac{1}{\Xi }\sum_{\varsigma =1}^{\Xi }(4\times \sigma_{\varsigma })$$
where the notation $\sigma_{\varsigma }=\mathrm{sqrt}\left(V\left({\hat{y}}_{\varsigma }^{\beta }\right)\right)$ represents the standard deviation of the prediction results at the ς-th dataset. The smaller the AoU, the higher the reliability of the model’s prediction will be. Figure 6a shows the validation result where the red dash line indicates the mean value of the estimation and the gray color represents the uncertainty of the estimation which is plotted by ${\hat{y}}_{\varsigma }^{v}\pm 2\times \sigma_{\varsigma }$. As can be seen, the MSE is 39.84 and the AoU is 0.12.

### 3.3. On-Line RUL Estimation

After the SAL-based feature selection model and GPR model were trained by using the data from T1 and T2, they were then applied to the testing dataset acquired from T3 with the aim of conducting the on-line RUL estimation. First, the dataset from T3 was used to generate a feature network dictionary where the size of the dictionary is 300 × 5670. Second, based on the off-line trained location Λ, the features of the test dataset were selected which can be expressed as $P_{s}^{\beta^{*}}$. Finally, the selected features were input into the GPR model for the RUL estimation. Figure 6b shows the predicted RUL on the test dataset producing an MSE of 192.65 and an AoU of 0.41.

### 3.4. Additional Experiment Designs

To validate our proposed RUL estimation framework, as can be seen in Table 1, two additional experimental designs denoted as E2 and E3 were carried out. In the case of E2, the training and validation datasets are from T1 and T3, and the testing dataset is from T2; while for E3, the datasets acquired from T2 and T3 were used as the training and validation datasets, and the dataset collected from T1 was utilized as the testing dataset. For both experiments, 7 features were selected from 5670 candidate features during the training and validation stage. Similar to E1, the selected features are also associated with AE and force signals only, with no dependence on vibration signals. The validation results of E2 and E3 were presented in Figure 7a and Figure 8a, respectively. Finally, after applying testing datasets to the trained SAL and GPR model, the RUL testing results are presented in Figure 7b and Figure 8b. Table 3 and Table 4 summarized the validation and testing results of E1, E2, and E3.

Furthermore, the parameter *l* is also set to 0 to evaluate the RUL predictions in order to highlight the advantages of the feature extension in the feature network dictionary. For E1, based on Equation (Equation 11), with a non-extended dictionary, the number of features in the dictionary is only 105 which contains less useful information than the extended feature network dictionary. After applying SAL, the number of selected features is 39, which corresponds to the whole 7 sensor outputs. Compared to Figure 6, which utilizes the extended feature network dictionary, the prediction results of the non-extended dictionary (see Figure 9) show an inferior performance and use more sensors. In the same way, E2 and E3 also show poorer performance when l=0. As a result, the extended feature network dictionary can potentially contain as much useful information as possible which can replace the installation of more sensors and incorporate more information meaning that the cost relating to sensor installation can be saved.

### 3.5. Comparative Study

Since our method has two parts containing the SAL-based feature selection method and GPR prediction method, some comparisons are made in terms of RUL predictions by using a Least Absolute Shrinkage and Selection Operator (LASSO) approach [27], a pure SAL algorithm, and a pure GPR algorithm:First, with respect to the LASSO approach, as can be seen in Table 3 and Table 4, the prediction results are promising and are close to that of our proposed method. However, the number of selected features exceeds 1000 in terms of E1, E2, and E3, which far exceeds the SAL-based feature selection method.Second, for the pure SAL algorithm, it can produce a very sparse solution with an adequate prediction performance (see Table 3 and Table 4), but it cannot output the uncertainty of the model which limits the ability of engineers/researchers to have an evaluation of the reliability of the prediction results.Last, for the pure GPR method, as presented in Table 3 and Table 4, it can output the uncertainty of the model, but the method is not sparse, i.e., it exploits the overall feature information for RUL prediction, which loses efficiency in high-dimensional spaces, especially when the number of features is tremendous.

As a result, the combination of SAL and GPR can take full advantage of the benefits of uncertainty estimation and overcome the drawbacks due to oversized features.

## 4. Case Study 2: NASA Ames Milling Datasets

To further demonstrate the effectivenss of the proposed RUL estimation framework, a second case study was conducted by analysing the NASA milling dataset [28]. Figure 10 depicts the setup of the experiment. As can be seen, AE sensors, vibration sensors, and motor current sensors are mounted to the table and the spindle of the Matsuura machining center MC-510V; moreover, six signal outputs which are represented as ${\mathrm{AE}}_{\mathrm{table}}$ (${\mathrm{AE}}_{\mathrm{table}}$ indicates AE signals at table), ${\mathrm{AE}}_{\mathrm{spindle}}$ (${\mathrm{AE}}_{\mathrm{spindle}}$ indicates AE signals at spindle), ${\mathrm{VB}}_{\mathrm{table}}$ (${\mathrm{VB}}_{\mathrm{table}}$ indicates table vibration signals), ${\mathrm{VB}}_{\mathrm{spindle}}$ (${\mathrm{VB}}_{\mathrm{spindle}}$ indicates spindle vibration signals), ${\mathrm{smc}}_{\mathrm{AC}}$ (${\mathrm{smc}}_{\mathrm{AC}}$ indicates AC spindle motor current), and ${\mathrm{smc}}_{\mathrm{DC}}$ (${\mathrm{smc}}_{\mathrm{DC}}$ indicates DC spindle motor current), respectively, are collected for estimating the RUL of the cutting tools. During the experiment, the cutting speed was set to 200 m/min, and two different depths of cut (DoC), namely, 1.5 mm and 0.75 mm, and two different feed rates, namely, 413 mm/min and 206.5 mm/min, were designed for the experiments. Furthermore, the workpieces for the experiments have two materials, which are cast iron and stainless steel J45 having the size of 483 mm × 178 mm × 51 mm. All experiments were performed a second time with the same parameters and a second cutting tool. The tools used for this case study are KC710-type carbide insets. Table 5 summarized the conducted experiments under different process parameter conditions. As can be seen from the table, it contains 8 pairs of experiments with the same 8 different parameter settings. For example, Experiment 9 is a repeat experiment of Experiment 1. Therefore, the dataset acquired from Experiment 1 can be considered as the training/validation dataset and the dataset collected from Experiment 9 can be used as the testing dataset, and this pair of experiments can be denoted as P1. Similarly, the rest of the datasets can be named in the same way. Table 6 summarizes these experimental pairs.

Figure 11a shows the RUL estimation results of the validation dataset of the experiment pair P1. During the off-line training/validation stage, the number of the candidate features in the feature network dictionary is 4185, and only 9 features related to 3 sensors, namely, ${\mathrm{AE}}_{\mathrm{table}}$, ${\mathrm{AE}}_{\mathrm{spindle}}$, and ${\mathrm{VB}}_{\mathrm{spindle}}$, were selected. This process can result in considerable cost savings in industrial applications because unselected sensors will not be used during the on-line test. After that, for the on-line estimation, as can be seen in Figure 11b, the testing results are consistent with the validation results in terms of MSE and AoU, which implies that our proposed RUL estimation framework inherits some robustness. Furthermore, the traditional LASSO approach, SAL approach, and GPR approach were also used to analyze the dataset for comparison with our method. As can be seen from the pictures shown in Figure 12, Figure 13 and Figure 14, they have higher MSE and AoU in comparison to our method. This means that the proposed framework has higher accuracy compared to other methods. Furthermore, Table 7 and Table 8 present the validation results and testing results of the 8 experiment pairs. As can be seen, LASSO can have promising results on validations datasets but has poor results on testing datasets meaning that this method is not robust. In terms of the proposed SAL and GPR method, for the majority of cases of the testing dataset, the performance of the proposed method has higher accuracy than the traditional methods in terms of the MSE and AoU demonstrating the superiority of our proposed RUL estimation framework over traditional techniques.

## 5. Discussion

To better understand the advantages of the proposed SAL and GPR-based RUL estimation framework, some discussions are summarized as follows:(1)Sparsity: This study proposed a novel feature network dictionary which can extend the number of candidate features under limited sensor conditions with the aim of containing as much useful information as possible. However, the features in the dictionary are usually redundant and may not be necessary for the RUL estimation of cutting tools. The proposed SAL feature selection method can reduce the number of candidate features and select the most significant features. Compared with the conventional feature selection method, such as LASSO, SAL has a very high sparsity and only a very few significant features can be selected.(2)Cost saving: For traditional feature selection methods, they have low sparsity. Therefore, they lead to the selection of features associated with more sensors. In contrast, the high sparsity of SAL leads to the selection of fewer sensors, which can save costs for online monitoring.(3)Limitations: The process of feature network dictionary extension can enlarge the number of candidate features but increase a significant amount of computation to select the features. Therefore, a more efficient optimization algorithm is needed to reduce the computational requirements, which will be studied in the future research.

## 6. Conclusions

In this paper, an innovative feature network dictionary was proposed to enlarge the number of candidate features under limited sensor conditions, and the developed dictionary can potentially contain as much useful information as possible. As demonstrated by the experiments, this process can replace the installation of more sensors and incorporate more information. After that, the Sparse Augmented Lagrangian (SAL) feature selection method was applied to select the significant features and the Gaussian Process regression (GPR) algorithm was applied to estimate the Remaining Useful Life (RUL) of cutting tools. The results from several case studies demonstrated the effectiveness of the proposed technique.

This study can be treated as the initial research for RUL estimation of cutting tools because it mainly focuses on simple machining processes and strategies. In future research activities, more experimental cases including different cutting processes and machining strategies can be tested to validate our proposed methods. Furthermore, we plan to use different materials in the training and testing stages for evaluating the RUL prediction results.

## Figures and Tables

**Figure 1 sensors-23-00413-f001:**
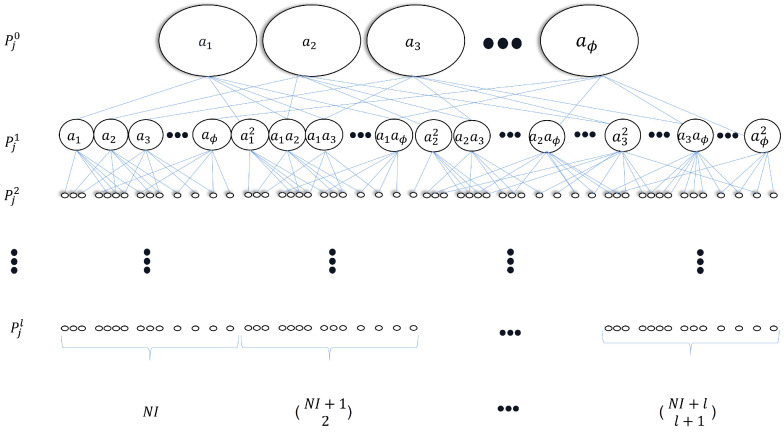
Process of generating the feature network dictionary during the *j*-th cutting process.

**Figure 2 sensors-23-00413-f002:**
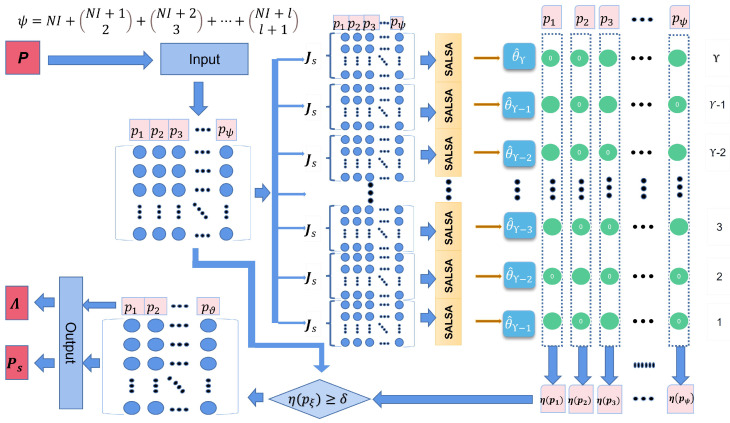
The implementation of SAL for feature selection.

**Figure 3 sensors-23-00413-f003:**
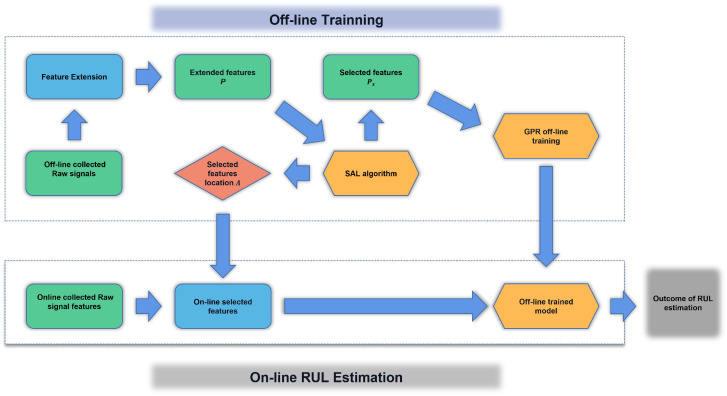
The process of off-line training and on-line RUL estimation of the Cutting Tool.

**Figure 4 sensors-23-00413-f004:**
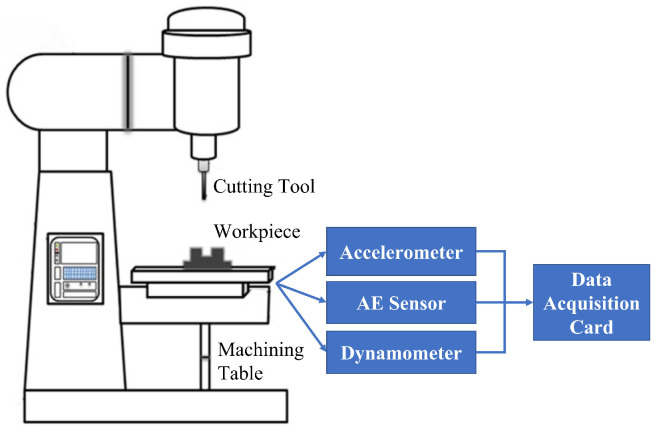
Illustration of the test rig.

**Figure 5 sensors-23-00413-f005:**
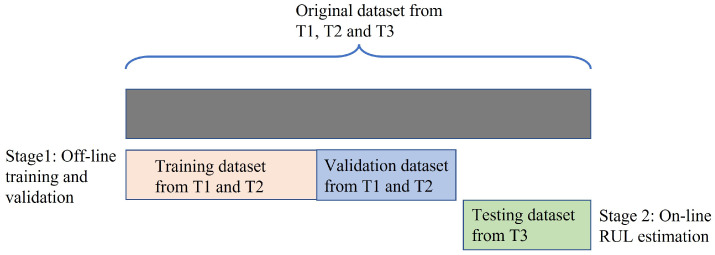
Design of the dataset partitioning of E1.

**Figure 6 sensors-23-00413-f006:**
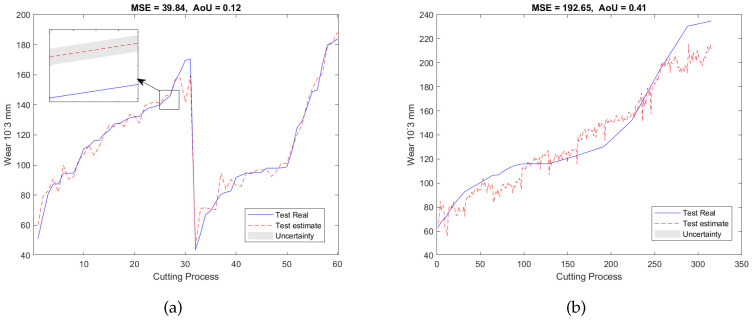
RUL prediction resultsof the experiment E1 of (**a**) the validation dataset and (**b**) the testing dataset.

**Figure 7 sensors-23-00413-f007:**
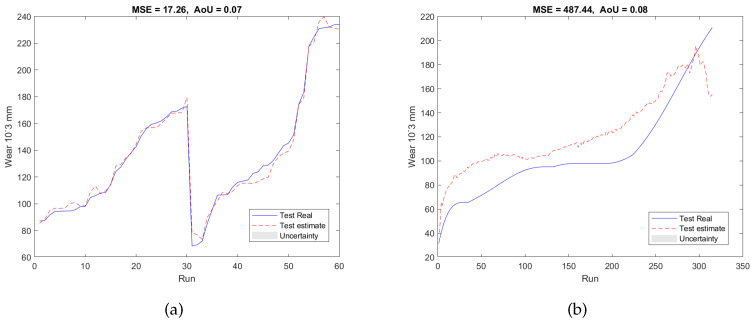
RUL predictionresults of the experiment E2 of (**a**) the validation dataset and (**b**) the testing dataset.

**Figure 8 sensors-23-00413-f008:**
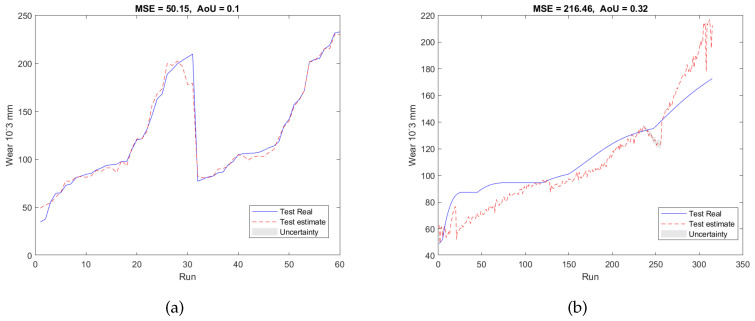
RUL predictionresults of the experiment E3 of (**a**) the validation dataset and (**b**) the testing dataset.

**Figure 9 sensors-23-00413-f009:**
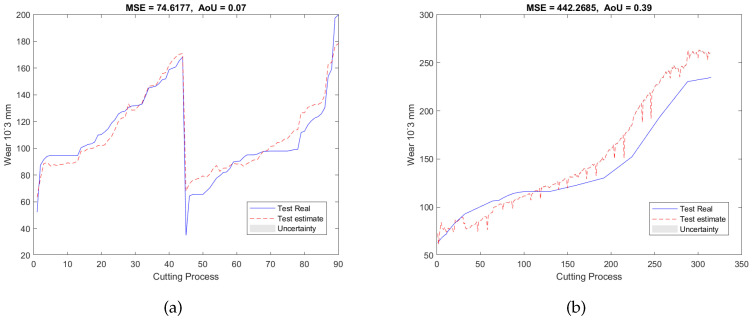
RUL predictionresults of the experiment E1 under the non-extended dictionary condition (l=0) of (**a**) the validation dataset and (**b**) the testing dataset.

**Figure 10 sensors-23-00413-f010:**
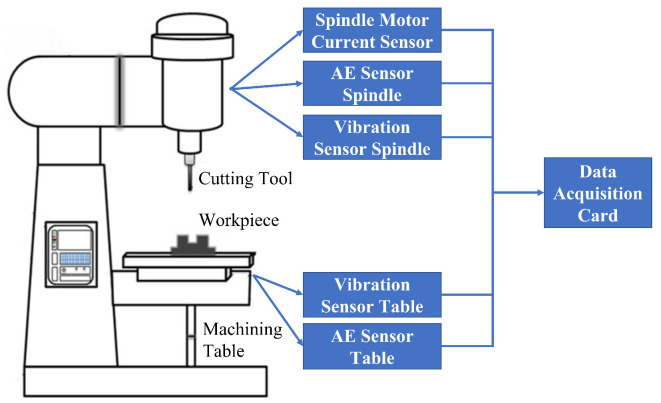
Experimental setup of the NASA Ames milling datasets [28].

**Figure 11 sensors-23-00413-f011:**
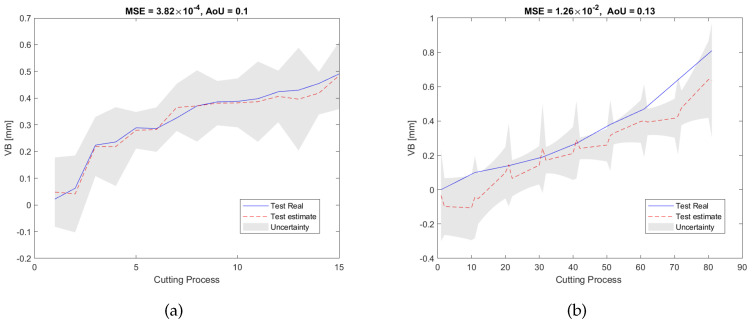
RUL prediction results of the proposed method of the experiment pair P1: (**a**) the validation dataset; (**b**) the testing dataset.

**Figure 12 sensors-23-00413-f012:**
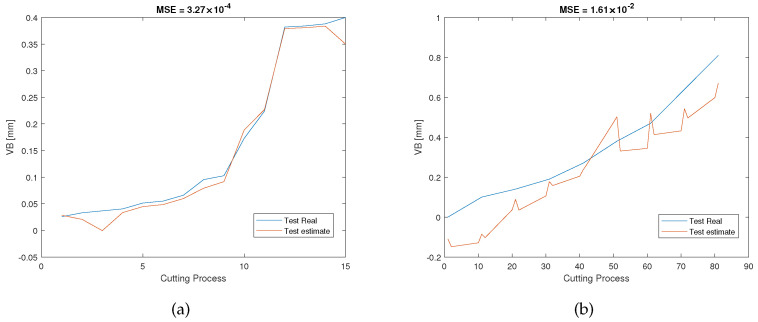
RUL prediction results of the LASSO method of the experiment pair P1: (**a**) the validation dataset; (**b**) the testing dataset.

**Figure 13 sensors-23-00413-f013:**
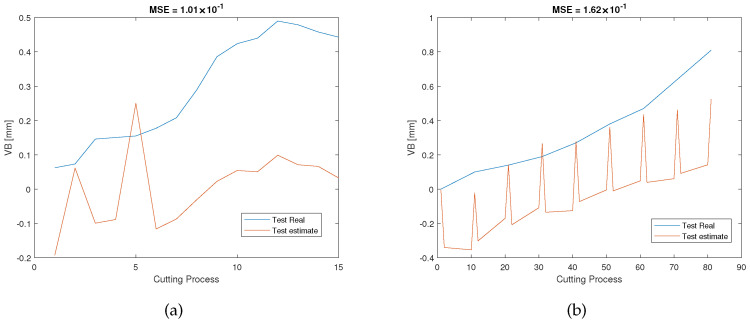
RUL prediction results of the SAL method of the experiment pair P1: (**a**) the validation dataset; (**b**) the testing dataset.

**Figure 14 sensors-23-00413-f014:**
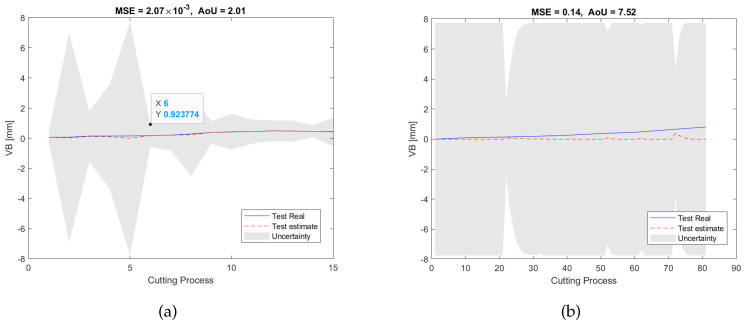
RUL prediction results of the GPR method of the experiment pair P1: (**a**) the validation dataset; (**b**) the testing dataset.

**Table 1 sensors-23-00413-t001:** Experimental design for data partitioning.

Symbol	Training/Validation	Testing
E1	T1 and T2	T3
E2	T1 and T3	T2
E3	T2 and T3	T1

**Table 2 sensors-23-00413-t002:** List of the utilized features.

Features	Expression
mean	$\mu =\frac{1}{n}\sum_{i=1}^{n}x$
variance	$\sigma^{2}=\frac{1}{n}\sum_{i=1}^{n}{\left(x-\mu \right)}^{2}$
standard deviation	$\sigma =\sqrt{\frac{1}{n}\sum_{i=1}^{n}{\left(x-\mu \right)}^{2}}$
skewness	$\gamma_{1}=\frac{\frac{1}{n}\sum_{i=1}^{n}{\left(x-\mu \right)}^{3}}{{\left(\frac{1}{n}\sum_{i=1}^{n}{\left(x-\mu \right)}^{2}\right)}^{\frac{3}{2}}}$
kurtosis	$K=\frac{\frac{1}{n}\sum_{i=1}^{n}{\left(x-\mu \right)}^{4}}{{\left(\frac{1}{n}\sum_{i=1}^{n}{\left(x-\mu \right)}^{2}\right)}^{2}}$
root mean square	$X_{rms}=\sqrt{\frac{\sum_{i=1}^{n}x^{2}}{n}}$
root mean square amplitude	$X_{r}={\left(\frac{1}{n}\sum_{i=1}^{n}\sqrt{\left|x\right|}\right)}^{2}$
rectification average	$X_{arv}=\frac{1}{n}\sum_{i=1}^{n}\left|x\right|$
peak to peak value	$X_{p}=x_{max}-x_{min}$
waveform factor	$k_{f}=\frac{X_{rms}}{X_{arv}}$
margin factor	$k_{f}=\frac{X_{max}}{X_{rms}}$
peak factor	$k_{a}=\frac{x_{p}}{x_{r}}$
median	$Q_{\frac{1}{2}}\left(x\right)=\left\{\begin{array}{cc} x_{\frac{n+1}{2}}^{{}^{\prime }}, & \;n\;is\;odd\\ \frac{x_{\frac{1}{2}n}^{{}^{\prime }}+x_{\frac{1}{2}n+1}^{{}^{\prime }}}{2}, & \;n\;is\;even\; \end{array}\right.$
maximum	$X_{max}=max\left(x\right)$
sum	$\mu =\sum_{i=1}^{n}x$

**Table 3 sensors-23-00413-t003:** Summary of the validation results of E1, E2, and E3.

Algorithms	MSE			AoU		
	E1	E2	E3	E1	E2	E3
SAL and GPR	**39.84**	**17.26**	**50.15**	**0.12**	**0.07**	0.10
LASSO	145.84	170.80	217.50	Non	Non	Non
SAL	123,45	89.74	67.31	Non	Non	Non
GPR	673.08	147.62	1192.31	**0.12**	5.15	**0.08**

**Table 4 sensors-23-00413-t004:** Summary of the testing results of E1, E2, and E3.

Algorithms	MSE			AoU		
	E1	E2	E3	E1	E2	E3
SAL and GPR	**192.65**	487.44	**216.46**	0.41	**0.08**	0.32
LASSO	2103.42	419.67	279.72	Non	Non	Non
SAL	458.65	254.36	222.79	Non	Non	Non
GPR	372.16	**202.71**	313.83	**0.15**	5.18	**0.08**

**Table 5 sensors-23-00413-t005:** Experimental conditions of NASA milling datasets.

Experiment	DoC	Feed	Material	Experiment	DoC	Feed	Material
1	1.5	0.5	castiron	9	1.5	0.5	castiron
2	0.75	0.5	casetsiron	10	1.5	0.25	castiron
3	0.75	0.25	castiron	11	0.75	0.25	castiron
4	1.5	0.25	castiron	12	0.75	0.5	castiron
5	1.5	0.5	steel	13	0.75	0.25	steel
6	1.5	0.25	steel	14	0.75	0.5	steel
7	0.75	0.25	steel	15	1.5	0.25	steel
8	0.75	0.5	steel	16	1.5	0.5	steel

**Table 6 sensors-23-00413-t006:** Experimental pairs of NASA milling datasets.

Symbol	Training/Validation	Testing
P1	Experiment 1	Experiment 9
P2	Experiment 2	Experiment 12
P3	Experiment 3	Experiment 11
P4	Experiment 4	Experiment 10
P5	Experiment 5	Experiment 16
P6	Experiment 6	Experiment 15
P7	Experiment 7	Experiment 13
P8	Experiment 8	Experiment 14

**Table 7 sensors-23-00413-t007:** Summary of the validation results of NASA milling datasets (**a**) MSE and (**b**) AoU.

(a)								
Algorithms		MSE × 10^−4^						
		E1	E2	E3	E4	E5	E7	E8
SAL and GPR		3.82	179.00	8.03	26.10	49.70	5.34	43.00
LASSO		**3.27**	**0.15**	**0.02**	**0.01**	**0.03**	**0.02**	**0.04**
SAL		1010	189.00	45.00	8.00	5.00	40.00	25.00
GPR		20.70	195.82	594.62	7.61	4.88	63.82	5.74
(**b**)								
Algorithms		AoU						
		E1	E2	E3	E4	E5	E7	E8
SAL and GPR		**0.10**	**0.06**	**0.16**	**0.71**	**0.53**	**0.21**	**0.16**
LASSO		Non	Non	Non	Non	Non	Non	Non
SAL		Non	Non	Non	Non	Non	Non	Non
GPR		2.01	**0.06**	0.17	1.87	1.52	2.64	1.73

**Table 8 sensors-23-00413-t008:** Summary of the testing results of NASA milling datasets, (**a**) MSE and (**b**) AoU.

(a)								
Algorithms		MSE × 10^−2^						
		E1	E2	E3	E4	E5	E7	E8
SAL and GPR		**1.26**	2.14	**3.36**	**0.74**	**1.02**	31.31	**3.08**
LASSO		1.61	2.26	3.73	1.14	1.27	**1.09**	4.99
SAL		16.20	68.72	2.21	0.78	1.89	10.90	5.30
GPR		14.00	**1.33**	7.91	11.83	3.07	28.02	6.88
(**b**)								
Algorithms		AoU						
		E1	E2	E3	E4	E5	E7	E8
SAL and GPR		**0.13**	0.47	**0.06**	**0.52**	0.26	**0.31**	**0.36**
LASSO		Non	Non	Non	Non	Non	Non	Non
SAL		Non	Non	Non	Non	Non	Non	Non
GPR		7.52	**0.06**	0.08	3.00	**0.13**	0.71	1.19

## Data Availability

The data presented in this study are available on request from the corresponding author.

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
