# Peer review of "Real-Time Remaining Useful Life Prediction of Cutting Tools Using Sparse Augmented Lagrangian Analysis and Gaussian Process Regression"

_sensors, 2022, doi:10.3390/s23010413_

Round 1

Reviewer 1 Report

1. Some Figures are low resolution, for example, Figure 4. Please replace it with a better one.

2. There is a typo in Figure 5, there are two Testing datasets. The first one is supposed to be the Training dataset. In addition, it would be better if Figure 5 is improved with the information for each stage: Training dataset, Validation dataset, and Testing dataset. 

3. The caption of Table 2 should be on top or above Table 2, not below Table 2.

4. What does "Expressio" means in Table 2?

5. Is Figure 10 from Ref. [27]? Authors need to have the approval to reuse the Figure.

6. Tables 7 and 8 are supposed to be positioned before the Conclusions. Typically, Conclusions do not have a Table.

Overall comment: the method used in the paper has been tested with real data. The result of the proposed method also has been compared with other methods.

Author Response

Please see my response in the attachment.

Reviewer 2 Report

1. The language should be improved to increase the readability of this paper.

2. The meaning of each symbol in the Equation should be given. Please check all the Equations to make sure the meaning of each symbol is given to increase the intelligibility of this paper.

3. Limitations and future work should be clarified in this paper.

4. A discussion section should be added to this paper.

5. The contribution of this paper is exaggerated, and the innovation of this paper is not clear.

Author Response

(The authors gave the same response as above.)

Reviewer 3 Report

The main objective of the article is to reduce costs for indirect methods of tool life prediction. A feature network dictionary is proposed for the first time for the indirect method, which can expand the number of candidate features under limited sensor conditions and the developed dictionary can potentially contain as much useful information as possible. A Sparse Augmented Lagrangian (SAL) feature selection method is proposed to reduce the number of candidate features and select the most important ones. During online monitoring, only selected sensors are used, and unselected sensors can be removed from the machine, and significant cost savings can also be achieved by this method. Through extensive experiments it can be seen that the output of the RUL estimation framework proposed in the paper outperforms the conventional methods, especially the cost savings of online RUL estimation. Therefore this paper can be published with minor modifications. The problem is shown as follows.

Chart format

(1) 16 lines of keywords followed by an extra parenthesis

(2) 104 lines In has an extra space in front of it

(3) 120 line 3 should be followed by an ellipsis instead of a comma

(4) The symbols marked by equation 8 are not on the same line as the equation

(5) 161 lines and should not be preceded by a comma

(6) 221, 248 lines of sentences are more to the right

(7) There is no uniformity in the punctuation of the figures as well as the tables that follow them, with some punctuation added and some not.

(8) There is a space before the 365 lines compared

(9) There is a problem with the indentation of the first line of 388 lines

(10) Who is marked 4 on line 397 for?

(11) 414 lines of paragraphs are not indented in the first line (you can indent paragraphs in the text to note the following)

(12) Table 7 Table 8 It is better to put (a) and (b) at the bottom of the table. Because the previous icon subscripts are below. And the two tables in Table 7 are not the same size.

(13) Figure 4(b) slightly lower definition

(14) 397, 387, 434 lines of the same natural paragraph is truncated by the chart, affecting the reading experience

Content

(15) Would it be too few to choose 8 from 5670 datasets and how can 8 datasets represent all the features?

(16) In line 337 MSE is 33.54 and AoU is 0.12. The value of MSE in figure (a) is 39.84. The values do not correspond.

(17) In line 379, for E1, E2 and E3, the number of selected features is more than 1000, which is much more than the SAL-based feature selection method. But in 3 it seems that the value of MSE of LASSO is greater than the value of MSE of SAL is worth, shouldn't the smaller the value of MSE the more accurate the model prediction? So how can this exceed be explained?

(18) Figure 12 Experimental RUL prediction results of the LASSO method for P1 (a) validation dataset (b) test dataset and Figure 13 Experimental RUL prediction results of the SAL method for P1 (a) validation dataset (b) test dataset. The value of AoU is not given therein.

Author Response

(The authors gave the same response as above.)

Round 2

Reviewer 1 Report

Dear Authors,

Thank you for your time and effort in revising the paper. I have checked the revised paper and I have no further comments.

Wish you all the best.

Kind regards,

- Reviewer 1 -

Reviewer 2 Report

This paper proposes a new feature network dictionary that can expand the number of candidate features under limited sensor conditions, and the developed dictionary can contain as much useful information as possible. This process can replace the installation of more sensors and contain more information. Then, the Sparse Augmented Lagrangian (SAL) feature selection method is proposed to reduce the number of candidate features and select the most significant features. Finally, the selected features are fed into a Gaussian process regression (GPR) model for RUL estimation.

1、 The processing object chosen for the test in the paper is aluminum material, does it have any effect on the accuracy of the subsequent prediction of the remaining useful life (RUL) of different materials.

2、 This paper does the relevant training and prediction based on sound signal and force signal, how different it is from the prediction based on sound signal, force signal and vibration signal, and further discusses the difference

Reviewer 3 Report

Agree
